# Cost-effectiveness of craniotomy versus decompressive craniectomy for UK patients with traumatic acute subdural haematoma

Sarah Pyne,[1] Garry Barton  ,[1] David Turner,[1] Harry Mee  ,[2] Barbara A Gregson,[3] Angelos G Kolias,[2,4] Carole Turner,[2,4] Hadie Adams,[2,4] Midhun Mohan,[2,4] Christopher Uff,[5] Shumaila Hasan,[5] Mark Wilson,[6] Diederik Oliver Bulters,[7] Ardalan Zolnourian,[7] Catherine McMahon,[8] Matthew G Stovell,[2] Yahia Al-Tamimi,[9,10] Simon Thomson,[11] Edoardo Viaroli,[2,4] Antonio Belli,[12] Andrew King,[13] Adel E Helmy,[2,4] Ivan Timofeev,[2,4] David Menon  ,[14] Peter John Hutchinson  ,[2,4] For the RESCUE-ASDH trial collaborators

**Correspondence to**
Dr Garry Barton;
g.barton@uea.ac.uk

## ABSTRACT

**Objective** To estimate the cost-effectiveness of craniotomy, compared with decompressive craniectomy (DC) in UK patients undergoing evacuation of acute subdural haematoma (ASDH).

**Design** Economic evaluation undertaken using health resource use and outcome data from the 12-month multicentre, pragmatic, parallel-group, randomised, Randomised Evaluation of Surgery with Craniectomy for Patients Undergoing Evacuation-ASDH trial.

**Setting** UK secondary care.

**Participants** 248 UK patients undergoing surgery for traumatic ASDH were randomised to craniotomy (N=126) or DC (N=122).

**Interventions** Surgical evacuation via craniotomy (bone flap replaced) or DC (bone flap left out with a view to replace later: cranioplasty surgery).

**Main outcome measures** In the base-case analysis, costs were estimated from a National Health Service and Personal Social Services perspective. Outcomes were assessed via the quality-adjusted life-years (QALY) derived from the EuroQoL 5-Dimension 5-Level questionnaire (cost-utility analysis) and the Extended Glasgow Outcome Scale (GOSE) (cost-effectiveness analysis). Multiple imputation and regression analyses were conducted to estimate the mean incremental cost and effect of craniotomy compared with DC. The most cost-effective option was selected, irrespective of the level of statistical significance as is argued by economists.

**Results** In the cost-utility analysis, the mean incremental cost of craniotomy compared with DC was estimated to be −£5520 (95% CI −£18 060 to £7020) with a mean QALY gain of 0.093 (95% CI 0.029 to 0.156). In the cost-effectiveness analysis, the mean incremental cost was estimated to be −£4536 (95% CI −£17 374 to £8301) with an OR of 1.682 (95% CI 0.995 to 2.842) for a favourable outcome on the GOSE.

**Conclusions** In a UK population with traumatic ASDH, craniotomy was estimated to be cost-effective compared with DC: craniotomy was estimated to have a lower mean cost, higher mean QALY gain and higher probability of a more favourable outcome on the GOSE (though not all estimated differences between the two approaches were statistically significant).

**Ethics** Ethical approval for the trial was obtained from the North West—Haydock Research Ethics Committee in the UK on 17 July 2014 (14/NW/1076).

**Trial registration number** ISRCTN87370545.

## STRENGTHS AND LIMITATIONS OF THIS STUDY

⇒ This study is based on individual patient-level data from a large, pragmatic, multicentre randomised trial.
⇒ It is both the first randomised trial and the first economic evaluation to compare craniotomy to decompressive craniectomy.
⇒ Multiple imputation was undertaken to account for missing data.
⇒ For ethical reasons, baseline EuroQoL 5-Dimension 5-Level scores were taken at discharge from the neurosurgical unit, rather than at randomisation.
⇒ A number of sensitivity analyses were undertaken to assess the robustness of conclusions to different assumptions in relation to these and other aspects.

## BACKGROUND

In the UK, an estimated 1.3 million people live with a traumatic brain injury-related disability and the annual societal cost has been estimated to be £15 billion (2015 cost levels).[1] Acute subdural haematoma (ASDH) is a common consequence[2] where craniotomy and decompressive craniectomy (DC) are the two mainstay treatments for surgical evacuation of the haematoma.[3] Both involve

the removal of a piece of the skull (bone flap) to evacuate the haematoma. With craniotomy, the bone flap is replaced, whereas with DC it is not. DC may help alleviate brain swelling and is undertaken with the view of a further operation being performed to rebuild the skull (cranioplasty). Craniotomy has the advantage that a patient will not need a later operation to rebuild the skull, but it may fail to control brain swelling in some patients. A systematic review found few studies comparing the two procedures, none of which were randomised, with contrasting evidence as to which was superior.[3]

Given this uncertainty as to whether craniotomy or DC is the more effective treatment for patients with ASDH, the choice of treatment is generally left to the discretion of the surgeon.[3] However, guidance/recommendations for the provision of different treatment options are now often based on estimated levels of cost-effectiveness.[4] Moreover, levels of cost-effectiveness may differ between these two surgical procedures as, for example, DC often requires cranial reconstruction using cranioplasty, which has additional costs and a significant complication profile[5] but may better alleviate brain swelling, translating into quality-of-life benefits.[5] Thus, here we report an economic evaluation[6] that was conducted alongside the RESCUE-ASDH (Randomised Evaluation of Surgery with Craniectomy for Patients Undergoing Evacuation of Acute Subdural Haematoma) trial,[5] to compare the cost-effectiveness of craniotomy vs DC for UK patients with traumatic ASDH.

## METHODS
### Participants
The RESCUE-ASDH trial[5] is a multicentre, international, pragmatic, parallel-group, randomised trial that compared craniotomy with DC. Patients were eligible if they were ≥16 years, had an ASDH on CT scan and the admitting surgeon felt that the haematoma needed evacuating either by craniotomy or DC. The economic evaluation was nested within the RESCUE-ASDH trial and based on UK participants only.

### Treatment and randomisation
Enrolled patients had their ASDH evacuated in the operating room under general anaesthesia. The bone flap was raised, the dura opened and the haematoma evacuated, after which patients were randomly assigned to receive either craniotomy (bone flap restored before skin closure) or DC (bone flap removed prior to skin closure with a view to being restored later). Patients were only randomised if either treatment was feasible, those patients whose brain was too swollen to allow replacement of the bone flap were not randomised. These patients would have the bone flap left out and were not included in the intention-to-treat (ITT) analysis presented in this paper. As a pragmatic study, management of patients preoperatively, intraoperatively and postoperatively was undertaken according to each centre's standard of care.

Blocked randomisation (block size 4) with 1:1 ratio was used, with allocation stratified by geographical region, age group, severity of injury and CT findings.[5] Patients randomised to craniotomy could have a DC at a later stage if their condition deteriorated and at the discretion of the treating clinician. It was not possible to blind patients, relatives and treating clinicians but the primary outcome (see below) was adjudicated centrally by blinded investigators.

### Measuring costs
Costs were estimated from a UK National Health Service (NHS) and Personal Social Services (PSS) perspective.[7] Resource use data were collected via two methods: hospital-recorded data and a Patient Self-Report (12-month follow-up) Questionnaire (PSRQ). Both methods of data collection were developed in consultation with hospital staff/patients and focused on big cost drivers/resources that were expected to differ between arms.[8] All resource use items that were costed (see below) were estimated in £ Sterling for the 2018/2019 financial year, resource use items undertaken for research purposes were not costed.

The hospital-recorded data included the following resource use items: details of the intervention (craniotomy or DC) including length of operation and graft details; time spent in the intensive care unit (ICU) and neurosurgical unit (NSU) during initial (index) admission; cranioplasties and shunt placements (these could be received as part of the index admission and/or after discharge from the NSU); any further neurosurgical procedures received during index admission.

The 12-month follow-up PSRQ could be completed by a relative/friend/carer if the participant was unable to complete it and referred only to the time since discharge from NSU. Information requested included the following resource use items: overnight stays in a hospital or other healthcare facility (length of stay, ward type, any associated skull/brain operation); healthcare professional visits (professional seen, frequency and most common location); head/brain scans (MRI, CT or 'other'); time in a care home; help received from a family member/friend or carer.

After assigning unit costs to the resource use items (see online supplemental table S1 for unit costs), the costs associated with both the hospital-recorded data and PSRQ resource use items, excluding wider societal costs (care home and help/carer costs), were summed to estimate the total NHS and PSS cost per participant. For each group, the mean total costs were estimated over the 12-month follow-up period, along with the associated p value for the mean cost difference between groups. An exception to the above was that, to avoid double-counting, patient self-reported postdischarge overnight stays with an associated skull/brain operation would not be costed if the total reported number was less than the total reported number of hospital-recorded postdischarge cranioplasty and shunt

procedures (including revisions). In line with previous work,[9] the higher of the two values was considered the most accurate. It should also be noted that patients who were known to have died postdischarge (mortality is collected as part of the primary outcome, see below) were not sent the PSRQ. As such, postdischarge costs for these participants would have been treated as missing and estimated via imputation (see below for details of the imputation methods, where time postdischarge was included in the multiple imputation (MI) model). In contrast, cost data for participants who died before discharge from their index admission would not have been considered missing as hospital-recorded data would still have been available for such participants (postdischarge costs were set as equal to zero for such participants).

### Measuring outcomes

To estimate health-related quality-of-life and conduct a cost–utility analysis,[6] in line with UK National Institute for Health and Care Excellence (NICE) guidance,[4 7] the EuroQoL 5-Dimension 5-Level questionnaire (EQ-5D-5L)[10] was combined with mortality data to estimate quality-adjusted life-year (QALY)[6] scores. Participants completed the EQ-5D-5L at discharge from NSU (assumed baseline score), 6 and 12 months follow-up (if discharged from NSU by these time points). As recommended at the time of analysis,[11] the cross-walk mapping function[12] was used to convert responses into utility scores (range: −0.594 (worse than death) to 1 (full health)). Participants who died were assigned a utility score of 0 on their date of death (death was collected as part of the hospital-recorded data as it was required for the primary outcome, see below). Utility values were used to estimate QALYs over 12 months, based on the total area under the curve method and linear interpolation.[13]

For ease of interpretation, as is convention,[14] the trial primary outcome measure, the (ordinal) extended Glasgow Outcome Scale (GOSE),[15–18] at 12 months, was converted into a binary scale using a fixed dichotomy analysis (ie, favourable vs unfavourable)[5] to enable a cost-effectiveness analysis[6] to be undertaken. Favourable outcomes were defined as upper severe disability or better while unfavourable outcomes included death, vegetative state and lower severe disability. A sliding dichotomy analysis[5] was also undertaken and is described in online supplemental appendix 2.

### Missing data

Missing data are common in randomised trials and can lead to bias and lack of precision.[19] As recommended, patterns of missing data were examined to explore the mechanism of missingness.[19] Accordingly, MI with chained equations (MICE) under missing at random was used to impute missing costs and outcome data, by treatment group. The 'mi impute chained' command (Stata V.17.0 (StataCorp)) was used to create 30 data sets (based on recommendations in relation to the level of missing data[19]) that were then pooled using Rubin's rules.[20]

Due to the way data were collected/different levels of missing data, missing data for costs were imputed for total index admission costs (hospital-recorded data collection), total cranioplasty and shunt costs (hospital-recorded data collection over 12-month trial period) and total postdischarge costs (patient self-report questionnaire data collection at 12 months). These three costs were then combined to estimate total NHS and PSS costs. For outcomes, missing data were imputed for utility scores (EQ-5D-5L) at baseline, 6 and 12 months, Glasgow Coma Scale (GCS) score[21 22] at baseline and GOSE score at 12 months. In addition to these costs and outcomes, the MI model also included age (years), sex and time postdischarge (the number of days from discharge to the 12-month point or death).

### Incremental analyses

For both the cost-utility and cost-effectiveness analyses, a 12-month within-trial, ITT approach was adopted. In this base-case analysis, patients were analysed according to the treatment to which they were randomised, regardless of the treatment received. No discounting was undertaken.

For the cost–utility analysis, to estimate the mean incremental cost and incremental effect (QALY gain) associated with craniotomy compared with DC, a seemingly unrelated regression (SUR) analysis was undertaken.[23] Regressions included those baseline variables expected to be predictive of total costs and outcomes: age (years), sex and baseline utility score. Assuming dominance,[6] where an intervention was both more costly and less effective, did not occur the incremental cost-effectiveness ratio (ICER=mean incremental cost/mean incremental QALY),[7] for craniotomy versus DC, would be estimated.[6] In the UK, NICE refers to a cost-effectiveness threshold of £20000–£30000 per QALY.[7] As such, if craniotomy had an ICER below this level, this would suggest it is cost-effective, compared with DC. It should be noted that economists have argued that decisions about treatment adoption should be made based on mean estimates, irrespective of whether such differences are statistically significant.[24] Therefore, the treatment option which is estimated to be most cost-effective should be provided.[25] This approach is consistent with the objective of maximising benefits from a given budget.

For the cost-effectiveness analysis, in terms of the incremental effect, the outcome (based on the GOSE) had a binary scale (favourable/unfavourable) and logistic regression[26] was undertaken to estimate the OR (95% CI) of a favourable outcome for craniotomy compared with DC. Separately, the mean incremental cost associated with craniotomy compared with DC was estimated using linear regression. Both regressions included variables age (years) and sex, which were expected to be predictive of total costs and GOSE outcomes. Together, in the absence of dominance, the incremental cost and incremental effect would enable the ICER to be estimated in terms of the cost per percentage increase in the odds of a favourable outcome.

## Decision uncertainty

To estimate the level of uncertainty associated with the decision,[19] the probability of craniotomy being cost-effective, compared with DC, at a threshold of £20 000/QALY on the cost-effectiveness acceptability curve (CEAC)[25] was calculated. This was estimated by combining the mean coefficients and covariance matrix from the SUR model, as described in Faria *et al.*[19] The CEAC was only estimated in relation to the cost–utility analysis.

## Sensitivity analyses

The above analysis constituted the base-case analysis[6] and was carried out in accordance with a prespecified health economic analysis plan (HEAP) (see: https://www.rescueasdh.org/trial-documents). To assess the robustness of conclusions, sensitivity analyses (SA) were undertaken.[6] To analyse the data from a wider cost perspective the care home and carer costs (which were excluded from the base-case analysis) were added to the total NHS and PSS costs (SA wider cost perspective). A further sensitivity analysis (for the cost-utility analysis only) tested the use of the EQ-5D-5L score at discharge from NSU as the baseline for QALY calculations. As any benefits could already have been partially/wholly achieved by discharge, QALY scores were re-estimated with the assumption that given the grave nature of the condition and following expert advice, participants had the lowest possible EQ-5D-5L score at baseline (date of index surgery): −0.594 (SA lowest EQ-5D-5L baseline score). Four further sensitivity analyses (including a per-protocol analysis) were conducted and are presented in online supplemental appendix 3. 'SA wider cost perspective' deviated from the HEAP, for reasons explained in online supplemental appendix 4.

## Patient and public involvement

The aforementioned patient self-report questionnaire was developed in consultation with non-trial patients.

## RESULTS

### Participants

Between September 2014 and April 2019, 248 UK patients were recruited, 126 in the craniotomy arm and 122 in the DC arm. Compared with the 450 patients recruited to the full (international) trial (the baseline characteristics of which are summarised in table 1 of Hutchinson *et al*[5]), these UK patients are slightly older (3.5 years on average) and more likely to be on antithrombotic medication (table 1).

Levels of missing data were slightly lower in the craniotomy group compared with the DC group for cost variables and outcome variables (except at baseline) (online supplemental table S2).

### Costs

Levels of resource use by intervention arm are summarised in table 2, under three main categories: (1) hospital-recorded index admission; (2) hospital-recorded cranioplasties and shunts and (3) patient-reported (PSRQ) postdischarge.

The hospital-recorded index admission data show that the length of stay in ICU and NSU was slightly lower in the craniotomy group compared with the DC group, but not significantly so. Only small numbers of other neurosurgical operations were reported. With regard to cranioplasties and shunts (index admission and postdischarge), as expected, more patients in the DC group had cranioplasties than in the craniotomy group (DC is prerequisite to a cranioplasty). There were, however, patients who were randomised to craniotomy that went on to have a DC, 21 of which had a cranioplasty in the 12-month follow-up period. Most cranioplasties use a synthetic material. Shunts were uncommon and occurred at a similar frequency between the groups. In terms of the patient-reported (PSRQ) postdischarge resource use, there were no significant differences between the groups for any of the parameters measured.

Mean cost estimates are summarised in table 3 and divided into the same three main categories. As

---

**Table 1** Baseline characteristics of UK patients

| Characteristics | Craniotomy (N=126) | DC (N=122) |
|---|---|---|
| Age (mean±SD)—years, n | 52.3±16.4, 126 | 51.7±15.9, 122 |
| Male sex—No./total n (%) | 96/126 (76.2) | 101/122 (82.8) |
| Any antithrombotic medication—No./n (%) | 21/115 (18.3) | 22/110 (20.0) |
| Presence of major extracranial injury requiring admission—No./n (%) | 66/123 (53.7) | 57/120 (47.5) |
| Glasgow Coma Scale (GCS) 3–8* | 85/120 (70.8) | 72/119 (60.5) |
| Initial CT brain findings | | |
| Presence of midline shift >5 mm—No./n (%) | 106/124 (85.5) | 105/121 (86.8) |
| Compression/absence of basal cisterns—No./n (%) | 101/124 (81.5) | 102/121 (84.3) |
| Presence of parenchymal contusions <25 cc—No./n (%) | 58/125 (46.4) | 60/121 (49.6) |

*A GCS score of 3–8 is defined as 'severe brain injury'.
DC, decompressive craniectomy; n, number of patients for whom data were available; N, number allocated to that trial arm; No., number of associated patients; SD, standard deviation.

**Table 2** Levels of resource use according to intervention arm over 12-month treatment period for all UK patients (based on available data)

| Resource use | Craniotomy (N=126) | DC (N=122) | P value* |
|---|---|---|---|
| **Hospital-recorded, index admission** | | | |
| Primary intervention received, not as randomised, No. | 13 (n=126) | 8 (n=122) | – |
| Duration of index surgery (hours), mean±SD (No./n) | 2.57±0.89 (122/122) | 2.50±0.93 (110/110) | 0.603 |
| ICU length of stay (index admission) (days), mean±SD (No./n) | 11.85±8.61 (123/126) | 13.52±11.28 (121/122) | 0.189 |
| NSU length of stay (index admission) (days), mean±SD (No./n) | 16.75±24.92 (93/122) | 21.30±31.10 (99/120) | 0.210 |
| Further DCs (index admission), No. | 15 (n=116) | 4 (n=116) | – |
| Further haematoma evacuations (index admission), No. | 9 (n=116) | 2 (n=116) | – |
| Further wound revisions (index admission), No. | 1 (n=116) | 6 (n=116) | – |
| Further other cranial operations (index admission)†, No. | 3 (n=116) | 2 (n=2) | – |
| **Hospital-recorded (cranioplasties and shunts (index admission and postdischarge)** | | | |
| Primary cranioplasties, No. | 21 (n=124) | 62 (n=121) | – |
| Cranioplasties requiring synthetic plate, No. (%) | 17 (81.0%) (n=21) | 46 (74.2%) (n=62) | – |
| Cranioplasty revisions, No. | 5 (n=124) | 7 (n=121) | – |
| Cranioplasties (primary/revisions) requiring readmission, No. | 17 (n=124) | 58 (n=121) | – |
| Primary shunts, No. | 5 (n=126) | 4 (n=118) | – |
| Shunt revisions, No. | 5 (n=126) | 2 (n=118) | – |
| Shunts (primary/revisions) requiring re-admission, No. | 4 (n=126) | 4 (n=118) | – |
| Postdischarge cranioplasty/shunt-related procedures (combined), No. | 21 (n=124) | 61 (n=118) | – |
| **Patient-reported, postdischarge** | | | |
| Overnight stay with associated skull/brain operation, No. | 13 (n=111) | 32 (n=95) | – |
| Any overnight stay excluding skull/brain operation, No. reporting ≥1 stay | 61 (n=111) | 54 (n=95) | – |
| Overnight stay on rehabilitation unit,‡ (days), mean±SD (No./n) | 32.51±63.09 (45/111) | 35.10±68.80 (36/90) | 0.782 |
| Overnight stay on NSU,‡ (days), mean±SD (No./n) | 0.49±2.00 (8/111) | 1.14±5.57 (5/95) | 0.252 |
| Overnight stay on ICU,‡ (days), mean±SD (No./n) | 0.13±1.33 (1/111) | 0.07±0.72 (1/95) | 0.731 |
| Overnight stay on other ward,‡ (days), mean±SD (No./n) | 4.94±18.08 (20/109) | 3.04±17.00 (10/93) | 0.447 |
| Healthcare professional contact, N reporting≥1 visit | 64 (n=109) | 47 (n=94) | – |
| Hospital doctor (visits), mean±SD (No./n) | 0.60±1.33 (28/106) | 0.61±1.46 (24/92) | 0.980 |
| Nurse (visits), mean±SD (No./n) | 2.20±16.53 (8/107) | 0.76±5.28 (7/92) | 0.426 |
| General practitioner (visits), mean±SD (No./n) | 1.23±2.44 (36/106) | 1.09±1.93 (30/93) | 0.656 |
| Physiotherapist (visits), mean±SD (No./n) | 2.38±7.11 (29/105) | 4.03±11.19 (19/91) | 0.213 |
| Occupational therapist (visits), mean±SD (No./n) | 1.56±3.41 (32/105) | 2.22±7.22 (19/92) | 0.407 |
| Speech therapist (visits), mean±SD (No./n) | 0.55±2.29 (10/107) | 0.31±1.40 (9/90) | 0.386 |
| Social worker (visits), mean±SD (No./n) | 0.16±0.77 (6/107) | 0.12±0.44 (7/92) | 0.665 |
| Community care assistant (visits), mean±SD (No./n) | 2.68±21.44 (3/106) | 2.84±20.45 (3/92) | 0.958 |
| Emergency department (visits), mean±SD (No./n) | 0.10±0.53 (5/107) | 0.18±0.61 (10/93) | 0.321 |
| Psychologist/neuropsychologist (visits), mean±SD (No./n) | 0.27±1.24 (7/107) | 0.46±2.72 (7/93) | 0.514 |
| Other healthcare professional (visits), mean±SD (No./n) | 0.03±0.22 (2/107) | 0.04±0.33 (2/93) | 0.699 |
| Head/brain scan, No. reporting ≥1 scan | 47 (n=111) | 44 (n=93) | – |
| MRI scans, mean±SD (No./n) | 0.31±0.62 (27/111) | 0.33±0.56 (28/93) | 0.745 |
| CT scans, mean±SD (No./n) | 0.33±0.67 (27/111) | 0.45±0.73 (33/93) | 0.228 |
| Other scans, mean±SD (No./n) | 0.04±0.19 (4/111) | 0.02±0.15 (2/93) | 0.543 |
| **Patient-reported, postdischarge (wider resource use)** | | | |
| Time in a care home (weeks), mean±SD (No./n) | 1.79±7.14 (10/109) | 3.53±10.40 (12/91) | 0.164 |
| Help from carer (hours), mean±SD (No./n) | 971±2017 (46/99) | 1000±2225 (36/86) | 0.925 |

*For the mean cost difference between groups,
†Excluding cranioplasties and shunts.
‡Excluding those reported (by the patient) to be associated with a skull/brain operation (estimates were instead based on hospital-recorded data (see online supplemental table S1).
DC, decompressive craniectomy; ICU, intensive care unit; n, number of patients for whom data were available; N, number allocated to that trial arm; No., number of patients in receipt of the resource item in question i.e. excluding zero values; NSU, neurosurgical unit; SD, standard deviation.

**Table 3** Estimates of mean cost (UK £ sterling, 2018/19) by treatment group over 12-month treatment period for all patients (based on available data)

| Cost component | Craniotomy (N=126) | DC (N=122) | P value* |
|---|---|---|---|
| **Hospital-recorded, index admission** | | | |
| Index neurosurgical procedure, mean cost±SD | 3648±1264 (n=122) | 3560±1315 (n=110) | 0.603 |
| Length of stay in NSU (index admission), mean cost±SD | 6109±9085 (n=122) | 7766±11 339 (n=120) | 0.210 |
| Length of stay in ICU (index admission), mean cost±SD | 20 039±14 566 (n=126) | 22 873±19 077 (n=122) | 0.189 |
| Further DCs (index admission), mean cost±SD† | 307±859 (n=116) | 82±536 (n=116) | 0.017 |
| Further haematoma evacuations (index admission), mean cost±SD | 165±638 (n=116) | 37±279 (n=116) | 0.048 |
| Further wound revision (index admission), mean cost±SD | 18±198 (n=116) | 110±551 (n=116) | 0.092 |
| Further other cranial operations (index admission),‡ mean cost±SD | 55±340 (n=116) | 37±279 (n=116) | 0.653 |
| **Total cost per patient (index admission), mean cost±SD** | **30 790±19 710 (n=109)** | **34 759±24 481 (n=102)** | **0.195** |
| **Hospital-recorded cranioplasties and shunts (index admission and postdischarge)** | | | |
| Cranioplasty procedures, mean cost±SD | 1059±2485 (n=124) | 3055±3352 (n=122) | <0.0001 |
| Shunt procedures, mean cost±SD | 212±1121 (n=126) | 150±834 (n=118) | 0.626 |
| Cranioplasty/shunt same day discount, mean cost±SD§ | −17±132 (n=124) | 0±0 (n=118) | 0.167 |
| **Total cost per patient (cranioplasties and shunts), mean cost±SD** | **1258±2983 (n=124)** | **3228±3677 (n=118)** | **<0.0001** |
| **Patient-reported, postdischarge** | | | |
| Overnight stays on rehabilitation unit, mean cost±SD¶ | 16 375±31 784 (n=111) | 17 677±34 660 (n=90) | 0.782 |
| Overnight stays on NSU, mean cost±SD¶ | 177±729 (n=111) | 415±2029 (n=95) | 0.252 |
| Overnight stays on ICU/HDU, mean cost±SD¶ | 213±2247 (n=111) | 125±1215 (n=95) | 0.731 |
| Overnight stays on 'other' ward, mean cost±SD¶ | 1746±6396 (n=109) | 1076±6015 (n=93) | 0.447 |
| All healthcare professional visits, mean cost±SD | 682±1108 (n=103) | 782±1578 (n=88) | 0.612 |
| All head/brain scans, mean cost±SD | 66±105 (n=111) | 93±101 (n=93) | 0.436 |
| **Total cost per patient (postdischarge PSRQ), mean cost±SD** | **19 699±34 193 (n=99)** | **17 948±32 183 (n=81)** | **0.726** |
| Time in a care home (wider perspective only), mean cost±SD | 3321±13 230 (n=109) | 6550±19 272 (n=91) | 0.164 |
| Carer time (wider perspective only), mean cost±SD | 16 762±34 828 (n=99) | 17 271±38 419 (n=86) | 0.925 |
| **Overall NHS and PSS cost per patient, mean cost±SD** | **48 509±46 934 (n=86)** | **53 573±47 092 (n=67)** | **0.510** |

*For the mean cost difference between groups.
†Based on mean duration of DC (from all index procedures) of 2.50 (n=110) hours for all randomised patients.
‡Excluding cranioplasties and shunts.
§A discount was applied to account for those shunt and cranioplasty procedures that occurred on the same day and were therefore assumed to be associated with a slightly shorter operation duration and NSU stay.
¶Overnight stays excluding those associated with a skull/brain operation.
DC, decompressive craniectomy; HDU, High Dependency Unit; ICU, intensive care unit; n, number of patients for whom data were available; N, number allocated to that trial arm; NHS, National Health Service; NSU, neurosurgical unit; PSRQ, Patient Self-Report Questionnaire; PSS, Personal Social Services; SD, standard deviation.

expected, given the procedure complexity and recovery time, total NHS and PSS costs are high in both groups. High index admission costs particularly accounted for this, largely due to the high cost of ICU stays, along with postdischarge costs, largely due to the high cost of overnight stays in a rehabilitation unit. There were, however, few significant differences between groups, the only notable one being the cost of cranioplasty procedures which, for the aforementioned reasons, was significantly higher in the DC group. As the number of postdischarge hospital-recorded cranioplasty/shunt procedures exceeded patient-reported overnight stays with an associated skull/brain operation (table 2), the latter has not been costed.

## Outcomes

Outcomes are summarised in table 4. Follow-up mean EQ-5D-5L scores were higher in the craniotomy group compared with the DC group, significantly so at 12 months. Furthermore, the change (increase) in EQ-5D-5L score from baseline was significantly higher at both 6 and 12 months in the craniotomy group compared with the DC group. There was no significant difference between groups for the total QALY score, based on available data.

At 12 months, the percentage of favourable GOSE scores was higher, but not significantly, in the craniotomy group compared with the DC group.

## Analyses
### Cost-utility analysis

For the base-case (based on ITT/MI), the mean difference in cost for the craniotomy group compared with the DC group was −£5520 (95% CI −£18 060 to £7020) with a mean QALY difference of 0.093 (95% CI 0.029 to 0.156) (table 5). Craniotomy, therefore, dominated DC; it was

**Table 4** Estimates of mean outcomes by treatment group over 12-month treatment period for all patients (based on available data)

| Item | Craniotomy (N=126) | DC (N=122) | P value* |
|---|---|---|---|
| Baseline EQ-5D-5L score, mean±SD | 0.260±0.353 (n=87) | 0.302±0.366 (n=91) | 0.441 |
| 6-month EQ-5D-5L score, mean±SD | 0.427±0.392 (n=102) | 0.370±0.393 (n=94) | 0.311 |
| 6-month change in EQ-5D-5L score, mean±SD | 0.184±0.345 (n=74) | 0.073±0.319 (n=71) | 0.046 |
| 12-month EQ-5D-5L score, mean±SD | 0.471±0.402 (n=111) | 0.336±0.414 (n=103) | 0.016 |
| 12-month change in EQ-5D-5L score, mean±SD | 0.218±0.367 (n=79) | 0.073±0.361 (n=78) | 0.013 |
| Total QALY score, mean±SD | 0.351±0.335 (n=68) | 0.338±0.366 (n=64) | 0.830 |
| 12-month GOSE score, % favourable† | 47.9 (n=121) | 37.4 (n=115) | 0.102 |

*For the mean difference between groups.
†Favourable for the GOSE score was defined as upper severe disability or better.
DC, decompressive craniectomy; EQ-5D-5L, EuroQoL 5-Dimension 5-Level; GOSE, Extended Glasgow Outcome Scale; n, number for whom data were available; N, number allocated to that trial arm; QALY, quality-adjusted life-year; SD, standard deviation.

estimated to be associated with both lower costs and more benefits.

## Cost-effectiveness analysis

For the craniotomy group compared with the DC group, the mean difference in cost was −£4536 (95% CI −£17 374 to £8301) with an OR of favourable outcome on the GOSE score of 1.682 (95% CI 0.995 to 2.842) (table 5). Again, craniotomy, therefore, dominated DC.

## Decision uncertainty

The base-case probability that craniotomy was cost-effective compared with DC, at a threshold of £20 000/QALY, was 87% (table 5). This indicates a high degree of certainty associated with the cost-utility analysis decision that craniotomy compared with DC is cost-effective at that threshold.

## Sensitivity analyses

In the sensitivity analyses, from a wider cost perspective and using the lowest EQ-5D-5L baseline score (for the cost−utility analysis only), craniotomy was again found to dominate DC (see table 5). Results of further sensitivity analyses, all of which are consistent with the base-case results, are presented in online supplemental table A1 (online supplemental appendix 3).

## DISCUSSION
### Main findings

In this UK population of patients with traumatic ASDH that warrants surgical evacuation, based on the results of the cost-utility and cost-effectiveness analyses, craniotomy dominated DC as it was estimated to have a lower mean cost, a higher mean QALY gain/higher probability of a more favourable outcome on the GOSE. Craniotomy was, therefore, estimated to be cost-effective, on the basis that the associated level of significance is considered to be irrelevant.[24 25] In the cost-utility analysis (QALY outcome), there was only an estimated 13% probability (at a threshold of £20 000/QALY) of making the wrong decision by choosing craniotomy. The results of the sensitivity analyses are in keeping with this result.

Within this study, it is important to highlight that costs were estimated from the viewpoint of the UK NHS and PSS and that associated resource use and outcome data were based only on participants from UK sites. As, for example, unit costs may differ outside the UK it is important to note that it is unclear whether these results are generalisable to sites outside the UK. Further associated research may, therefore, be warranted in relation to this and that ≥20% of patients who were randomised to craniotomy went on to have a DC (as an ITT approach

**Table 5** Estimates of the mean incremental cost, incremental effect (QALY gain or OR) and cost-effectiveness of craniotomy compared with DC in the base case and two sensitivity analyses (based on imputed data)

| Cost-utility analysis | Incremental cost (95% CI) (N=126) | QALY gain (95% CI) (N=122) | ICER | CEAC* |
|---|---|---|---|---|
| Base-case: imputed | −£5520 (−£18 060 to £7020) | 0.093 (0.029 to 0.156) | Dominant | 87% |
| SA wider cost perspective | −£17 793 (−£34 658 to −£928) | 0.094 (0.030 to 0.159) | Dominant | 99% |
| SA lowest EQ-5D-5L baseline score | −£5445 (−£17 547 to £6658) | 0.089 (0.025 to 0.152) | Dominant | 87% |
| **Cost-effectiveness analysis** | **Incremental cost (95% CI)** | **OR (95% CI)†** | **ICER** | |
| Base case | −£4536 (−£17 374 to £8301) | 1.682 (0.995 to 2.842) | Dominant | – |
| SA wider cost perspective | −£16 900 (−£33 807 to £7) | 1.693 (0.998 to 2.871) | Dominant | – |

*Probability of being cost-effective on the CEAC at a threshold of £20 000 per QALY.
†For a favourable outcome for craniotomy compared with DC, based on the Extended Glasgow Outcome Scale, as described in the Methods section.
CEAC, cost-effectiveness acceptability curve; 95% CI, 95% confidence interval; DC, decompressive craniectomy; Dominant, lower mean costs and higher mean effect; EQ-5D-5L, EuroQoL 5-Dimension 5-Level; ICER, incremental cost-effectiveness ratio; N, number allocated to that trial arm and included in the analysis – imputation was undertaken as part of all presented analyses; QALY, quality-adjusted life-years; SA, sensitivity analysis.

was adopted these patients were included in the craniotomy arm in the base-case analysis).

## Strengths and limitations

Regarding health-related quality of life, QALY scores (EQ-5D-5L recorded at all time points) were available for 53% of participants only and the amount of missing data was greater at discharge than at 6 and 12 months (online supplemental table S2). Some missing EQ-5D-5L baseline (NSU discharge) data may be due to participants being discharged at short notice or at the weekend when a research nurse was not available. As some patients had not yet been discharged from the hospital by 6 months, this may explain the higher rates of EQ-5D-5L missing data at this time point compared with 12 months. Postdischarge costs (based on patient self-report data) were also missing for 27.4% of patients at 12 months (online supplemental table S2). Such missing data are a limitation, but we did impute missing data and take an ITT approach, which meant that all patients were still included in the analysis.

A further limitation is that, for ethical reasons, baseline EQ-5D-5L scores were taken at discharge from NSU, rather than at randomisation. Therefore, any benefits could be underestimated by assuming this score is the baseline score. To test the potential impact of this, a sensitivity analysis (SA lowest EQ-5D-5L baseline score) assumed the baseline EQ-5D-5L to be that of the worst possible health state (−0.594). The results differ little from the base case (table 5) with craniotomy still dominating DC. It should also be noted that, in the cost-effectiveness analysis, as the cost and outcome regressions are performed separately any correlation between the cost and outcome variables would not be accounted for. A final limitation is that the 12-month follow-up period may not be sufficient to capture all expected cranioplasties. For example, of those randomised to DC (122), only 62 had received a cranioplasty within the 12 month follow-up period. As such, further cranioplasties (aside from those who were randomised but did not receive DC (8/122) and those who had died (31/122)) could take place beyond the 12-month period. Though this is a limitation, the inclusion of such costs would only be expected to increase the long-term incremental cost of DC, and therefore, not change the conclusion that craniotomy dominated DC.

## Comparisons with other studies

We are not aware of any previous economic evaluations that have specifically compared craniotomy with DC for patients with ASDH. Previous economic evaluations of DC have been undertaken,[27–31] but these have had different comparators and used a variety of different populations/methods (most developed a decision analytical model to estimate costs and benefits,[29–31] and the two papers[27 28] that used actual patient data were not based on randomised data and were of a smaller sample size than used here, with different cost perspectives and timeframes). Thus, it is difficult to make direct comparisons to our study, and the use of different methods may explain

why there were differences in the results as to whether DC was estimated to be cost-effective or not.[27–31]

## Implications

In a UK population of patients with traumatic ASDH, craniotomy was estimated to have a lower mean cost, a higher mean QALY gain and a higher probability of a more favourable outcome on the GOSE, dominating DC. Based on the QALY, there was a high probability that craniotomy, compared with DC, was cost-effective (at a threshold of £20 000/QALY). When sensitivity analyses were conducted, the main conclusion (that craniotomy was, therefore, estimated to be cost-effective) remained unchanged. Consequently, the health economic analysis supports the recommendation, based on the primary outcome,[5] that a craniotomy should be undertaken, rather than a DC, if it is operatively feasible to replace the bone flap.

**Author affiliations**
[1]Norwich Medical School, University of East Anglia, Norwich, UK
[2]Department of Clinical Neurosciences, University of Cambridge, Cambridge, UK
[3]Neurosurgical Trials Group, Wolfson Research Centre, Newcastle University, Newcastle upon Tyne, UK
[4]Division of Neurosurgery, Addenbrooke's Hospital, Cambridge, UK
[5]The Royal London Hospital, London, UK
[6]Department of Neurosurgery, St Mary's Hospital, London, UK
[7]University Hospital Southampton NHS Foundation Trust, Southampton, UK
[8]Department of Neurosurgery, The Walton Centre NHS Foundation Trust, Liverpool, UK
[9]Department of Neurosurgery, Sheffield Teaching Hospitals NHS Foundation Trust, Sheffield, UK
[10]Academic Directorate of Neurosciences, Sheffield Teaching Hospitals NHS Foundation Trust, Sheffield, UK
[11]Department of Neurosurgery, Leeds General Infirmary, Leeds, UK
[12]Department of Neurosurgery, University Hospitals Birmingham NHS Foundation Trust, Birmingham, UK
[13]Department of Neurosurgery, Salford Royal Hospital Manchester Centre for Clinical Neurosciences, Salford, UK
[14]Division of Anaesthesia, University of Cambridge, Cambridge, UK

**Collaborators** RESCUE-ASDH trial collaborators (to be indexed on PubMed): Harry Mee, Angelos G Kolias, Carole Turner, Hadie Adams, Midhun Mohan, Edoardo Viaroli, Adel E Helmy, Ivan S Timofeev, David K Menon, Peter J Hutchinson – University of Cambridge. Peter J. Kirkpatrick, Hadie Adams, Nicola Owen, Kirsty Grieve, Alexis Joannides, Ellie Edlmann, Karen Caldwell, Silvia Tarantino, Francesca Hill, Liz Corteen, Janet Corn, Matthew G Stovell, Mathew Guilfoyle, Mohammad Naushahi, Richard Mair, Ibrahim Jalloh, Mark Kotter, Kamila Walker, Tamara Tajsic, Sara Venturini, Selma Tülü – Cambridge University Hospital, Cambridge, UK. Christopher Uff, Shumaila Hasan, Chipo Chitsenega, Geetha Boyapati – Royal London Hospital, London, UK. Muhammad Bhatti, Malik Zaben, Belinda Gunning, Natalia Ermalai, Dmitri Shastin, Mutwakil Abdulla, Liudmila, Joseph Merola – University Hospital of Wales, Cardiff, UK. Stuart Smith, Laurence Glanz, Ashwin Kumaria, Lani Patterson – Queen's Medical Centre, Nottingham, UK. Antonio Belli – 1. NIHR Surgical Reconstruction and Microbiology Research Centre, Birmingham, UK. 2. Institute of Inflammation and Ageing, University of Birmingham, Birmingham, UK. Colin Bergin, Maximina Ventura, Emma Toman – NIHR Surgical Reconstruction and Microbiology Research Centre, Birmingham, UK. Giles Critchley, Laura Ortiz-Ruiz de Gordoa, Prasanna Epaliyanage, Husam Georges – Royal Sussex County Hospital, Brighton, UK. Peter Whitfield, Sam Jeffrey, Charlotte Eglinton, Natasha Wilmhurst – Derriford, Hospital, Plymouth, UK. Manjunath Prasad, Shahid Khan, Philip Kane, Emanuel Cirstea – James Cook University Hospital, Middlesbrough, UK. Catherine McMahon, Geraint Sunderland, John Kitchen, Rushid Zakaria – The Walton Centre, Liverpool, UK. Simon Thomson, Soumya Mukherjee, Ian Anderson, Linetty Makawa, Mary Kambafwile – Leeds General Infirmary, Leeds, UK. Marios C. Papadopoulos, Mathew Joseph Gallagher, Siobhan Kearney, Sarah Trippier – St. George's Hospital,

London – UK. Mark Wilson, Hani Marcus, Rhys Thomas, Sonia Fernandez Lopez, Louise Young, Andrea D'Mello, Jo-Anna Conyngham – St. Mary's Hospital, London, UK. Diederik Bulters, Ardalan Zolnourian, Patrick Holton, Miriam Taylor, Charlaine Reeve – Southampton University Hospital, Southampton, UK. Jonathan Pollock, Ciaran Hill, Elisa Visentin, Laura Parker – Queen's Hospital, Romford, UK. Kismet Hossain-Ibrahim, Paul Johnston – Ninewells Hospital, Dundee, UK. Vasileios Arzoglou, Emma Clarkson, Arif Zafar, Efosa Ukponmwan – Hull Royal Infirmary, Hull, UK. Christos Tolias, Joseph Frantzias, Anastasios Giamouriadis – King's College London, London, UK. Yahia Al-Tamimi, Adam Wahba, Patrick Easton, Rose Clegg, Grace Cole – Royal Hallamshire Hospital, Sheffield, UK. Damian Holliman, Louise Finlay – Royal Victoria Infirmary, Newcastle, UK. Daniel Holsgrove, Alex Leggate, Jane Perez, Louise Harrison – Salford Royal Hospital, Manchester, UK. Gareth Roberts, Terrie-Louise Cromie, Marianne Hare, Sonia Raj – Royal Preston Hospital, Preston, UK. Tim Lawrence, Ruichong Ma, Javier Magan Ventura, Neil Davidson – John Radcliffe Hospital, Oxford, UK. Richard Nelson, Ruth Womer, Beverley Fulkner – Frenchay Hospital/North Bristol NHS Trust, Bristol, UK. Shabin Joshi, Geraldine Ward – Walsgrave General Hospital, Coventry, UK. Ioannis Fouyas, Julie Woodfield, Himanshu Shekhar, Kareen Damley, Emma Fleming – Western General Hospital, Edinburgh, UK. Roddy O'Kane, Michael Canty – Queen Elizabeth University Hospital, Glasgow, UK. Indira Devi Bhagavatula, Dhananjaya Ishwar Bhat, Dhaval Prem Shukla, Kanti Konar, Nagesh Shanhag, Vaishali NI Valluri – National Institute of Mental Health and Neurosciences (NIMHANS), Bangalore, India. Manoj Kumar Tewari, Manjul Tripathi – Post Graduate Institute of Medical Education and Research (PGIMER), Chandigarh, India. Deepak Gupta, Ashish Bindra, Kaveri Sharma – All India Institute of Medical Sciences (AIIMS), Delhi, India. Mathew Joseph – Christian Medical College, Vellore, India. Nicole Keong, Christine Lock, Chen Min Wei, Julian Han, Janell Kwok, Nicolas Kon Kam King – National Neuroscience Institute, Singapore. Martina Stippler, Emmalin Nelton – Beth Israel Medical Center, USA. Louis Anthony Whitworth, Sonja Stutzman, Christopher Madden, Caryn Harper, Tracey Moore, Vin Shen Ban, Alice Salazar – The University of Texas Southwestern Medical Center, USA. Rocco Armonda, Jason J. Chang, George B. Moses, Patricia Tanjucto – MedStar Washington Hospital Center, Washington, USA. Jamie Ullman, Orseola Arapi, Betsy Moclair – Northwell Shore University Hospital (Northwell Health), New York, USA. Nrupen Baxi, Erin Lewis, Melvin Stone – Jacobi Medical Center and Montefiore Medical Center, New York, USA. John Adair Prall, Meghan Baldwin, Jamie Jones – Littleton Adventist Hospital (Centura Health), Colorado, USA. Clare Gallagher, Ish Bains – Foothills Medical Centre, Calgary, Canada. Kesava Reddy, Paula Carroll – Hamilton Health Sciences, Ontario, Canada. Leodante Da Costa, Nadia Scantlebury – Sunnybrook Hospital, University of Toronto, Toronto, Canada. Fahad Alkherayf, Rafael Ochoa Sanchez, Ioana Moldovan – The Ottawa Hospital, Ottawa, Canada. Sean Christie, Lisa Julien – Queen Elizabet II Health Sciences Centre, Halifax, Canada. Kostas Fountas, Thanasis Paschalis – University Hospital of Larissa, Medical School (University of Thessaly), Larissa, Greece. Sandro Krieg, Sebastian Ille – Technische Universität München, Munich, Germany. Maria Luisa Gandia Gomez – Hospital Universitario La Paz, Madrid, Spain. Alfonso Legares, Ana Maria Castaño Leon – Hospital Universitario 12 de Octubre, Madrid, Spain. András Büki, Gábor Lenzsér – University of Pécs, Pécs, Hungary. Tariq Khan, Mukhtar Khan – Northwest General Hospital and Research Center, Peshawar, Pakistan. Franco Servadei, Massimo Tomei – Humanitas Research Hospital, Milan, Italy. Vairavan Narayanan, Ronie Romelean Jayapalan – University of Malaya Medical Centre, Kuala Lumpur, Malaysia. Martin Hunn, Panagiota Gkolia, Emily Galea ¬– The Alfred Hospital, Melbourne, Australia. Sarah C. Pyne, Garry R. Barton, David A Turner – Health Economics Group, Norwich Medical School, University of East Anglia. Barbara Gregson – Neurosurgical Trials Group, Wolfson Research Centre, Newcastle University, Newcastle upon Tyne. Trial co-applicants and/or protocol contributors (to be indexed on PubMed): Peter J. Hutchinson, Garry R. Barton, Clare Gallagher, Paula Kareclas, Angelos G. Kolias, Peter J. Kirkpatrick Sridevi Nagarajan, David Turner, James Piercy, Peter Kirkpatrick, A. David Mendelow, David K. Menon, Andrew King, Antonio Belli, John D. Pickard, Paul Brennan, Christopher Cowie, Cambridge Clinical Trials Unit staff (to be indexed on PubMed): Carol Davis-Wilkie, Tapiwa Tungamirai, Kerstin Wolf, Kamila Walker, Natalia Igosheva, Alicia Gore, Paula Kareclas, Michele Jillings, Christopher Bushell Trial Steering Committee members (to be indexed on PubMed): Anthony Bell (chair), Allison Hirst, Peter McCabe, Simon Shaw Independent Data Monitoring and Ethics Committee members (to be indexed on PubMed): Martin Smith (chair), Joan Grieve, Jonathan Cook.

**Contributors** GB, DT, HM, BAG, AGK, CT, HA, MM, CM, AB, AGK, DM and PJH contributed to the conception/design of the work. SP, GB, DT, HM, BAG, AGK, CT, HA, MM, CU, SH, MW, DOB, AZ, CM, MGS, YA-T, ST, EV, AEH, IT, DM and PJH contributed to the acquisition of data. SP, GB and DT conducted the analysis. All authors contributed to the interpretation of data/drafting of the paper (led by SP and GB) and approved the final manuscript. Guarantor: SP

**Funding** This project was supported by the Health Technology Assessment (HTA) Programme (project number 12/35/57) and will be published in full in the HTA journal at https://fundingawards.nihr.ac.uk/award/12/35/57; The RESCUE-ASDH trial is an 'embedded study' linked with the CENTER-TBI project (https://www.center-tbi.eu/) of the European Brain Injury Consortium. CENTER-TBI was a large-scale collaborative project, supported by the FP7 Program of the European Union (grant number 602150); RESCUE-ASDH ISRCTN Registry number, ISRCTN87370545. Study protocol is available at https://fundingawards.nihr.ac.uk/award/12/35/57. We thank the patients who participated in the RESCUE-ASDH trial, their families, and all the collaborating clinicians and research staff, and we thank the staff of the Cambridge Clinical Trials Unit for their support.

**Disclaimer** The views expressed are those of the authors and not necessarily those of the NHS, the National Institute for Health and Care Research (NIHR), or the Department of Health and Social Care.

**Competing interests** No support from any organisation other than the National Institute for Health and Care Research was received for the submitted work. BAG has received consulting fees from Cambridge University Hospitals NHS Foundation Trust. AGK is supported by a Senior Lectureship at the School of Clinical Medicine, University of Cambridge, the Wellcome Trust, and the Royal College of Surgeons of England. MW has received support for attending meetings and/or travel for presentations with the Wilderness Medical Society and Royal College of Surgeons of Edinburgh, is a member of the Trauma Clinical Reference group for the NHS, meetings secretary for the Society of British Neurosurgeons and a non-salaried medical director of GoodSAM. PJH is supported by a Research professorship and Senior Investigator award from the NIHR, the NIHR Cambridge Biomedical Research Centre and the Royal College of Surgeons of England.

**Patient and public involvement** Patients and/or the public were involved in the design, or conduct, or reporting, or dissemination plans of this research. Refer to the Methods section for further details.

**Patient consent for publication** Not applicable.

**Ethics approval** This study involves human participants and was approved by North West—Haydock Research Ethics Committee, UK. 17 July 2014 (14/NW/1076). Due to the life-threatening nature of the condition, the operation to remove an ASDH is undertaken as soon as possible after admission to a hospital with neurosurgical services on site. These patients will be incapacitated due to the head injury, and therefore, unable to give consent for trial entry themselves. If the next of kin is known/available they will be asked to give agreement to the patient entering the trial but often they are not present or cannot be traced or there is no time for a discussion with them. In these cases the operation to evacuate an ASDH will need to go ahead as it is a matter of life or death for the patient and the next of kin will be traced and informed after the operation. Patients who are incapable of giving consent in emergency situations are an established exception to the general rule of informed consent in clinical trials. This is clearly acknowledged in the Declaration of Helsinki. If participants regain capacity while in the hospital they will be given information about the clinical trial and their consent will be sought to continue in the trial. (The above was taken from the Trial protocol that was approved by the NHS Research Ethics Committee).

**Provenance and peer review** Not commissioned; externally peer reviewed.

**Data availability statement** Data are available on reasonable request. Reasonable requests to make relevant anonymised participant level data available will be considered by the trial team. The trial protocol (reference 15) and a prespecified health economic analysis plan (HEAP) (see: https://www.rescueasdh.org/trial-documents) are also available.

**ORCID iDs**
Garry Barton http://orcid.org/0000-0001-9040-011X
Harry Mee http://orcid.org/0000-0002-1314-3962
David Menon http://orcid.org/0000-0002-3228-9692
Peter John Hutchinson http://orcid.org/0000-0002-2796-1835

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
