## [Reviewer comments · BMJ Open]

ARTICLE DETAILS

TITLE (PROVISIONAL)	Cost-effectiveness of craniotomy versus decompressive craniectomy, for UK patients with traumatic acute subdural hematoma
AUTHORS	Pyne, Sarah; Barton, Garry; Turner, David; Mee, Harry; Gregson, Barbara; Koliass, Angelos; Turner, Carole; Adams, Hadie; Mohan, Midhun; Uff, Christopher; Hasan, Shumaila; Wilson, Mark; Bulter, Diederik; Zolnourian, Ardan; McMahon, Catherine; Stovell, Matthew G; Al-Tamimi, YAHIA; Thomson, Simon; Viaroli, Edoardo; Belli, Antonio; King, Andrew; Helmy, Adel E; Timofeev, Ivan; Menon, David; Hutchinson, Peter; RESCUE-ASDH, trial collaborators

VERSION 1 – REVIEW

REVIEWER	Dixon, Pdraig Oxford University
REVIEW RETURNED	27-Feb-2024

GENERAL COMMENTS	This study assessed the cost-effectiveness of craniotomy versus decompressive craniectomy for UK patients with acute subdural hematoma (ASDH). Data from the RESCUE-ASDH trial were used for this economic evaluation. The study included 248 patients randomized to either craniotomy or decompressive craniectomy. Costs were analyzed from the perspective of the NHS and personal social services, while outcomes were measured using Quality-Adjusted Life Years (QALYs) and the Extended Glasgow Outcome Scale (GOSE). The results indicated that craniotomy was cost-effective, showing lower mean costs, higher mean QALY gain, and a higher probability of a favorable outcome on the GOSE. Overall, this is a well-reported and well-conducted within-trial economic evaluation and publication is merited. The inclusion of a completed CHEERS 2022 checklist and the use of a HEAP to guide analysis and reporting is commendable. COMMENTS P5 “Component costs” - components of what? I assume this refers to the cost items enumerated in the previous paragraph – suggest redrafting this P6 “the extended Glasgow Outcome Scale (GOSE) at 12 months was collapsed into a fixed dichotomy analysis” - I had to look at the trial protocol to confirm that this scale is ordinal rather than
--

	cardinal. I would justify the use of a binary treatment of this scale by noting that any analysis of fixed interval between units on the continuous scale would be uninterpretable. I would also reference the text in the protocol regarding recommendations and conventions regarding analysis of this outcome. P6 “30 data sets (in line with the level of missing data” - Reference 19 recommends as least as many imputations are used for $(100) * (\% \text{ of missing data})$ and would be worth clarifying your drafting. Sentence is also missing a closing parenthesis. P6 “(ICER = mean incremental cost/QALY),” If you’re going to define costs as incremental in the numerator, then you should also define the QALY as incremental in the denominator P6 “Therefore, the treatment option which is estimated to be most cost-effective should be provided, regardless of the associated level of uncertainty.” This wording is slightly misleading – the intervention that is more likely to be cost-effective should be recommended, but this isn’t completely independent from uncertainties, notwithstanding that “statistical significance” shouldn’t enter into this conclusion in any event (viz. the Claxton reference 24). I would delete the second clause of the quoted sentence. I would also avoid this phrasing in the Discussion “...(craniotomy dominated DC as it was estimated to have a lower mean cost, a higher mean QALY gain /higher probability of a more favourable outcome on the GOSE (though none of these estimated differences were statistically significant).” since this isn’t relevant and I didn’t see indications that the trial wasn’t powered on the economic outcomes in any event. Likewise, this drafting is unclear (“irrelevant” intended instead of “irrespective”) and could be deleted from the Discussion “on the basis that the associated level of significance is considered to be irrespective” P7 – I’m not clear how the binary cost-effectiveness analysis was undertaken “As the outcome is binary (favourable/unfavourable), logistic (logit) regression was undertaken to estimate the odds ratio (95% CI) of a favourable outcome for craniotomy compared with DC. Mean incremental costs associated with craniotomy compared with DC were estimated using linear regression. Both regressions included variables age (years) and sex, which were expected to be predictive of total costs and GOSE outcomes.” You use SUR for the continuous QALY outcomes, presumably because it is reasonable to exploit the joint information in the error terms of the cost and QALY equations. Why not do the same here? Were the equations estimated separately and then means compared in the QALY? If so, the uncertainty around each estimate won’t be correct (by the logic of the SUR approach being correct) and this will impact CEACs etc. I suggest noting this as a limitation. Note also that the <code>–suest –</code> command in Stata can be used combine logistic and linear regressions in SUR-type models in Stata. I suspect this won’t have a major impact on any of your conclusions, but perhaps worth quickly confirming. P7 – suggest deleting reference to logit here “logistic (logit) regression” P7 – Fieller’s theorem – I’m uncertain how Fieller’s theorem entered these calculations – was it used in both the continuous
--	---

	and binary outcomes, and if so, how? I haven't been able to follow how cost-effectiveness probabilities/CEACs were calculated from Fieller's theorem – mentioned on p7 and in the results, but not otherwise. P10 – you mention outlying costs – is there any evidence that this was an issue? Could easily be checked rather than offering speculation P10 – how were deaths recorded? Presumably in trial case reports? P11 “ The main strength of this economic evaluation is that it is based on a large, multi-centre, randomised trial.” Not sure this is a strength – the analysis is based on a UK substudy of an international trial, with results from elsewhere not entering into the economic analysis, which reads more like a limitation. Indeed, the discussion in the Results section notes further research is warranted on non UK economic conclusions.
--	---

REVIEWER	Agus, Ashley Northern Ireland Clinical Trials Unit, Health Economics
REVIEW RETURNED	15-Mar-2024

GENERAL COMMENTS	Thank you for inviting me to review this interesting paper on the cost-effectiveness of craniotomy versus decompressive craniotomy. It is generally well written and will be a good addition to the health economic literature in this field. I did have a number of comments and suggestions which I hope the authors find useful. Background: The background section would benefit from more information on why clinicians would choose one method over the other in practice. It reads as if either could be performed, but in reality the bone flap cannot always be replaced. So some information about the risks/benefits of craniotomy (with bone flap replaced) would be beneficial here as these are only provided for decompressive craniotomy. Methods: The methods used for the economic evaluation are in line with the NICE guidance for health technology appraisals and the CHEERS checklist confirms the key information has been included. EQ-5D-5L should be used consistently throughout (not EQ-5D). Participants who died were assigned a utility score of 0 on their date of death, which is acceptable, how were costs handled for these participants? A unit cost per hour of theatre time is used to cost the different procedures based on length of surgery- was the unit cost specific to the neurosurgery? Unfortunately I wasn't able to find the publication from the reference provided [Public Health Scotland. Theatre - direct cost per hour by speciality. Edinburgh, 2019]. Multiple imputation methods were used to impute missing data, but could the authors clarify which variables were imputed and which were in the imputation model please? I'm not convinced that odds-ratios are the best outcome to use for the CEA as they are not easily interpreted within the CEA/ ICER framework. The method section would benefit from explaining how you were would obtain a cost per favourable outcome from the
---

	odds ratio. Alternatively could the outcome be analysed as the proportion/number of favourable outcomes instead? Results: The authors should explain what the “full analysis population” is, and how the sample used in the economic evaluation was selected in the methods section. They reference the main results paper (Hutchinson et al, 2023) but it’s not clear in that paper either. Table 2- It would be helpful if the authors provided the % of participants using the difference resource items alongside the counts / means. Only half (62/121) of the participants in the DC arm had received a cranioplasty within 12 months. Presumably these participants will go on to receive it at some point- so was the time horizon not long enough to capture these? Is this a limitation of the analysis? Would cranioplasties in the DC group occur as readmissions? If so, why are there not at least 62 cranioplasties (primary/revisions) requiring re-admission for DC group rather than just 58? Table 2 has a footnote “§Combines ‘Currently in paid/unpaid work’ with ‘hours working per week (paid or unpaid)’ and ‘reported return-to-work date’ to estimate mean hours worked per participant in 12 month follow-up period.” But I don’t think this was considered in the analysis. In Table 3 & 4 the top row states C has n=126 and DC has n=122, but this is misleading as the tables are based on available data. This should be stated in the title. Table 5 are all the results in this table on imputed data (not just base-case?). If so it would be useful to put this in the title and the n=126, and n=122 can be placed next to the appropriate treatment arm on the top row. It would then be helpful to provide the total costs and outcomes by treatment arm as well as the differences. Table 5 also just states Odds-Ratio- is this an incremental difference? It’s not clear. Discussion: Reference to Table 5 should be removed. The sentence “Craniotomy was therefore estimated to be cost-effective, on the basis that the associated level of significance is considered to be irrespective”. This should read Craniotomy compared to decompressive craniotomy was therefore.... I believe the authors mean “irrelevant” rather than “irrespective”.
--	--

VERSION 1 – AUTHOR RESPONSE

Reviewer: 1

Dr. Padraig Dixon, Oxford University

Comments to the Author:

This study assessed the cost-effectiveness of craniotomy versus decompressive craniectomy for UK patients with acute subdural hematoma (ASDH). Data from the RESCUE-ASDH trial were used for this economic evaluation. The study included 248 patients randomized to either craniotomy or decompressive craniectomy. Costs were analyzed from the perspective of the NHS and personal social services, while outcomes were measured using Quality-Adjusted Life Years (QALYs) and the Extended Glasgow Outcome Scale (GOSE). The results indicated that craniotomy was cost-effective, showing lower mean costs, higher mean QALY gain, and a higher probability of a favorable outcome on the GOSE.

Overall, this is a well-reported and well-conducted within-trial economic evaluation and publication is

merited. The inclusion of a completed CHEERS 2022 checklist and the use of a HEAP to guide analysis and reporting is commendable.

We thank the reviewer for their comments

COMMENTS

P5 “Component costs” - components of what? I assume this refers to the cost items enumerated in the previous paragraph – suggest redrafting this

The reviewer was correct in their assumption. In response to this comment, we have specifically listed the hospital-recorded and PSRQ resource use items that were costed and refer to ‘The costs associated with both the aforementioned hospital-recorded and PSRQ resource use items...’ rather than using the term ‘Component costs...’.

P6 “the extended Glasgow Outcome Scale (GOSE) at 12 months was collapsed into a fixed dichotomy analysis” - I had to look at the trial protocol to confirm that this scale is ordinal rather than cardinal. I would justify the use of a binary treatment of this scale by noting that any analysis of fixed interval between units on the continuous scale would be uninterpretable. I would also reference the text in the protocol regarding recommendations and conventions regarding analysis of this outcome.

In line with this comment, we now clarify that, for ease of interpretation, and as is convention, a fixed dichotomy analysis was undertaken in order to translate the ordinal scale into a binary scale: unfavourable versus favourable outcome. We have also included the suggested reference.

P6 “30 data sets (in line with the level of missing data” - Reference 19 recommends as least as many imputations are used for $(100) * (\% \text{ of missing data})$ and would be worth clarifying your drafting. Sentence is also missing a closing parenthesis.

We have added a parenthesis, checked the number of data sets, and updated this section to clarify 30 data sets is in line with the recommendations made in reference 19.

P6 “(ICER = mean incremental cost/QALY),” If you’re going to define costs as incremental in the numerator, then you should also define the QALY as incremental in the denominator

We have implemented this suggestion

P6 “Therefore, the treatment option which is estimated to be most cost-effective should be provided, regardless of the associated level of uncertainty.” This wording is slightly misleading – the intervention that is more likely to be cost-effective should be recommended, but this isn’t completely independent from uncertainties, notwithstanding that “statistical significance” shouldn’t enter into this conclusion in any event (viz. the Claxton reference 24). I would delete the second clause of the quoted sentence.

We have implemented this suggestion

I would also avoid this phrasing in the Discussion “...(craniotomy dominated DC as it was estimated to have a lower mean cost, a higher mean QALY gain /higher probability of a more favourable outcome on the GOSE (though none of these estimated differences were statistically significant).” since this isn’t relevant and I didn’t see indications that the trial wasn’t powered on the economic outcomes in any event.

As suggested, we have removed the reference to the differences not being statistically significant (see the 'Main Findings' section of the Discussion).

Likewise, this drafting is unclear ("irrelevant" intended instead of "irrespective") and could be deleted from the Discussion "on the basis that the associated level of significance is considered to be irrespective"

We thank the reviewer for picking up this error, we have now changed 'irrespective' to 'irrelevant'

P7 – I'm not clear how the binary cost-effectiveness analysis was undertaken "As the outcome is binary (favourable/unfavourable), logistic (logit) regression was undertaken to estimate the odds ratio (95% CI) of a favourable outcome for craniotomy compared with DC. Mean incremental costs associated with craniotomy compared with DC were estimated using linear regression. Both regressions included variables age (years) and sex, which were expected to be predictive of total costs and GOSE outcomes."

You use SUR for the continuous QALY outcomes, presumably because it is reasonable to exploit the joint information in the error terms of the cost and QALY equations. Why not do the same here? Were the equations estimated separately and then means compared in the QALY? If so, the uncertainty around each estimate won't be correct (by the logic of the SUR approach being correct) and this will impact CEACs etc. I suggest noting this as a limitation. Note also that the `–suest –` command in Stata can be used combine logistic and linear regressions in SUR-type models in Stata. I suspect this won't have a major impact on any of your conclusions, but perhaps worth quickly confirming.

Public Health Scotland. Scottish health service costs. Costs Book 19 (April 2018 to March 2019). See file R142X, available on the archive web-site (accessed April 5th 2024): <https://webarchive.nrscotland.gov.uk/20231203022410/https://www.isdscotland.org/Health%2DTopics/Finance/Costs/File-Listings-2019.asp>

Multiple imputation methods were used to impute missing data, but could the authors clarify which variables were imputed and which were in the imputation model please?

We have revised the second paragraph of the 'Missing data' section of the Methods to include this information:

For costs missing data was imputed at the level of total costs for: index admission, cranioplasty and shunt, and post-discharge. For outcomes missing data was imputed for utility scores (EQ-5D-5L) at baseline, 6 and 12 months, Glasgow Coma Scale (GCS) score^{21,22} at baseline and GOSE score at 12 months. In addition to these costs and outcomes, the MI model also included age (years), sex and time post-discharge (the number of days from discharge to the 12-month point or death).

I'm not convinced that odds-ratios are the best outcome to use for the CEA as they are not easily interpreted within the CEA/ ICER framework. The method section would benefit from explaining how you would obtain a cost per favourable outcome from the odds ratio. Alternatively could the outcome be analysed as the proportion/number of favourable outcomes instead?

To help with the interpretation of the CEA we have reworded this paragraph in the Methods to explain how the incremental cost, incremental effect and ICER would be estimated. A reference which includes a chapter on the interpretation of the logistic regression models has also been inserted.

Hosmer, DW. Lemeshow, S. Sturdivant, RX. Applied Logistic Regression (3rd edition). Wiley 2013, Hoboken, New Jersey (Reference 27).

Results:

The authors should explain what the "full analysis population" is, and how the sample used in the

economic evaluation was selected in the methods section. They reference the main results paper (Hutchinson et al, 2023) but it's not clear in that paper either.

Within the 'Participants' section of the Methods for this paper we explain that RESCUE-ASDH⁵ was a multicentre international randomised trial and that the economic evaluation was based on RESCUE-ASDH UK trial participants only. In light of this comment, in the 'Participants' section of the Results for this paper, we have corrected a typographical error (452 has been changed to 450) and explain that the 450 patients, to which we compare the 228 participants in the economic evaluation to, are those that were recruited to the full (international) trial (the baseline characteristics of which are summarised in Table 1 of Hutchinson et al. ⁵).

Would cranioplasties in the DC group occur as readmissions? If so, why are there not at least 62 cranioplasties (primary/revisions) requiring re-admission for DC group rather than just 58?

It is possible for a participant to receive both a DC and cranioplasty as part of their index admission i.e. without a re-admission. The following revised statement explains that this is a possibility "...cranioplasties and shunt placements (these could be received as part of the index admission and/or after discharge from the NSU)..." (see Measuring costs section of the Methods).

Table 2 has a footnote "§Combines 'Currently in paid/unpaid work' with 'hours working per week (paid or unpaid)' and 'reported return-to-work date' to estimate mean hours worked per participant in 12 month follow-up period." But I don't think this was considered in the analysis.

As stated by the reviewer, this was not considered in the analysis within this paper, it has therefore been deleted.

In Table 3 & 4 the top row states C has n=126 and DC has n=122, but this is misleading as the tables are based on available data. This should be stated in the title.

As suggested, we have clarified (in the Table titles) that the data presented within Tables 3 and 4 is based on available data. N is now also defined in the Table footnotes.

Table 5 are all the results in this table on imputed data (not just base-case?). If so it would be useful to put this in the title and the n=126, and n=122 can be placed next to the appropriate treatment arm on the top row. It would then be helpful to provide the total costs and outcomes by treatment arm as well as the differences.

Yes, all the results presented in this Table are based on imputed data. In line with this suggestion, we have clarified this in the Table title and now state N=126 and N=122 in the top row. We have also denoted in the footnote that N represents the number allocated to that trial arm and included in the analysis. We have not provided the total costs and outcomes as these are already reported (for available data in Tables 3 and 4, respectively).

Table 5 also just states Odds-Ratio- is this an incremental difference? It's not clear.

We have now defined the Odds Ratio in the Table Footnote

Discussion: Reference to Table 5 should be removed. The sentence "Craniotomy was therefore estimated to be cost-effective, on the basis that the associated level of significance is considered to be irrespective". This should read Craniotomy compared to decompressive craniotomy was therefore.... I believe the authors mean "irrelevant" rather than "irrespective".

As suggested, we have removed the reference to Table 5 and replaced “irrespective” with “irrelevant”.

VERSION 2 – REVIEW

REVIEWER	Dixon, Padraig Oxford University
REVIEW RETURNED	18-Apr-2024

GENERAL COMMENTS	Thank you for the responses and updates to the paper. Two small comments First, consider altering the first sentence under the "Decision uncertainty" heading to clarify that coefficients and covariance matrices are taken from the SUR model (if I've interpreted that statement correctly eg "estimates of the mean coefficients and covariance matrix from the SUR model were combined..." The drafting of this entire sentence is also unwieldy - could perhaps break into more than one sentence and make them all simpler and shorter. Second, I don't understand the new paragraph on missing data. Was the same imputation model used for both cost and non-cost outcomes (as would typically be recommended)? I don't follow how missing data were imputed at the "at the level of total costs for: index admission, cranioplasty and shunt, and post-discharge". Does this mean costs were calculated separately for these elements or total cost was calculated using these data?
---

VERSION 2 – AUTHOR RESPONSE

Reviewer: 1

Dr. Padraig Dixon, Oxford University

Comments to the Author:

Thank you for the responses and updates to the paper. Two small comments

First, consider altering the first sentence under the "Decision uncertainty" heading to clarify that coefficients and covariance matrices are taken from the SUR model (if I've interpreted that statement correctly eg "estimates of the mean coefficients and covariance matrix from the SUR model were combined..." The drafting of this entire sentence is also unwieldy - could perhaps break into more than one sentence and make them all simpler and shorter.

As suggested, we have amended the existing text, inserting reference to the SUR model, which the reviewer has understood correctly. We have also inserted the acronym 'SUR' at first reference to 'seemingly unrelated regression' in the 'incremental analysis' section. We agree that the sentence had become too long and obtuse. This has now been restructured into two, clearer sentences.

Second, I don't understand the new paragraph on missing data. Was the same imputation model used for both cost and non-cost outcomes (as would typically be recommended)? I don't follow how missing data were imputed at the "at the level of total costs for: index admission, cranioplasty and shunt, and post-discharge". Does this mean costs were calculated separately for these elements or total cost was calculated using these data?

Yes, the same imputation model was used for both costs and non-cost outcomes and we have made this clearer by adding 'costs and outcomes' to the existing sentence: '...MAR was used to impute missing costs and outcome data...'

In terms of the methods of imputation, missing data for costs was imputed for total index admission costs, total cranioplasty and shunt costs, and total post-discharge (patient-reported) costs. These three costs were then combined to estimate total NHS and PSS costs. This approach was adopted due to the way the data-collection methods were structured/different levels of missing data. Index admission costs were collected using hospital-recorded data at the time of discharge, cranioplasty and shunt data were collected using hospital-recorded data over the whole 12-month trial period, and other post-discharge costs were recorded using a patient self-report questionnaire at 12-months. More details on this are given in 'Missing data' section. In response to this comment, we have now re-worded this paragraph to make it clearer how and why missing data for costs was imputed.

In addition to the above comments we noticed that one reference appeared twice in the reference list (references 7 and 10 were the same and the latter has therefore been deleted). Accordingly, the references in the paper have been updated as well.